# Bipartite network models to design combination therapies in acute myeloid leukaemia

Mohieddin Jafari [1✉], Mehdi Mirzaie[1], Jie Bao[1], Farnaz Barneh[2], Shuyu Zheng [1], Johanna Eriksson[1], Caroline A. Heckman [3] & Jing Tang[1✉]

Combination therapy is preferred over single-targeted monotherapies for cancer treatment due to its efficiency and safety. However, identifying effective drug combinations costs time and resources. We propose a method for identifying potential drug combinations by bipartite network modelling of patient-related drug response data, specifically the Beat AML dataset. The median of cell viability is used as a drug potency measurement to reconstruct a weighted bipartite network, model drug-biological sample interactions, and find the clusters of nodes inside two projected networks. Then, the clustering results are leveraged to discover effective multi-targeted drug combinations, which are also supported by more evidence using GDSC and ALMANAC databases. The potency and synergy levels of selective drug combinations are corroborated against monotherapy in three cell lines for acute myeloid leukaemia in vitro. In this study, we introduce a nominal data mining approach to improving acute myeloid leukaemia treatment through combinatorial therapy.

[1] Research Program in Systems Oncology, Faculty of Medicine, University of Helsinki, Helsinki, Finland. [2] Prinses Maxima Center for Pediatric Oncology, 3584 CS Utrecht, Utrech, the Netherlands. [3] Institute for Molecular Medicine Finland - FIMM, HiLIFE - Helsinki Institute of Life Science, iCAN Digital Precision Cancer Medicine Flagship, University of Helsinki, Helsinki, Finland. ✉email: mohieddin.jafari@helsinki.fi; jing.tang@helsinki.fi

Studies on cases with advanced cancers have shown that less than 10% of patients have actionable mutations, and the improvement of outcomes is unobserved in a randomised trial of precision medicine based on genomic profiles[1]. The current limitation of genomics-centric personalised medicine falls short of the enormous heterogeneity and lack of actionable and sustainable treatment options. With a few exceptions, patient genomic signatures with clinical pathology do not typically predict drug responses. More precisely, cancer can principally be considered a signalling disease, not a genetic disease. There is a wealth of data that has validated this hypothesis, including signalling behaviours involved in growth factor and nutrient responses, the process of entering and exiting the cell cycle, ensuring that chromosomes are segregated in an orderly, efficient and accurate manner during mitosis, and apoptosis[2,3]. On the other hand, the complexity of crosstalk between signalling pathways necessitates to modify multiple targets in cancer cells; otherwise, a lack of complete response, resistance, and relapse will emerge during the course of treatment.

Despite the fact that large amounts of small molecules or drugs have been tested on many cell lines or patient-derived samples, using single drugs as monotherapies to cure cancer might not be a promising strategy, as it is known that the complex interactions of various biochemical components can induce drug resistance during the treatment of cancer[4–6]. As a matter of fact, monotherapy, the slogan of *one target one drug*, is inefficient in curing complex diseases, such as cancer[7,8]. Combination therapy or polytherapy with synergistic drugs may achieve a more effective and safer outcome by targeting several targets in the same or separate pathways of the complex system[4,9]. To better identify the synergistic drug combination based on precision medicine, we need ex vivo drug screening to decipher the functional impact of cancer genomics at the phenotypic level and to understand their interactions in the context of biological networks[10–12]. Therefore, understanding network biology may provide a unique opportunity to leverage the rich source of drug response data to offer network-based models for combinatorial therapy. These network models have shown promise for developing clinical decision support tools to discriminate functional patient subclasses[13,14]. Even though there are networks reconstructed to model biological mechanisms of diseases and predict drug combination synergies based on molecular data[15–18], network models have not been systematically applied to patient data, such as the drug response data of patient-derived samples, to predict patient-customised drug combinations[14]. Instead, the ex vivo drug response data are straightforwardly translated into the clinic for patient treatment since these individualised experiments represent the efficiency of some approved drugs on patient-derived primary cultures[19,20].

In 2018, the Beat AML programme reported a cohort of 672 tumour specimens collected from 531 patients, analysing the ex vivo sensitivity for 122 drugs alongside the mutational status and the gene expression signatures of the samples[21]. Despite the dearth of large patient-related drug response datasets, some large cell line–based datasets, such as genomics of drug sensitivity in cancer (GDSC) and ALMANAC, can offer a strong source of supporting evidence for predictions. The GDSC database contains the responses of 1001 cancer cell lines to 265 anti-cancer drugs, providing a rich source of information to connect genotypes with cellular phenotypes and to identify cancer-specific therapeutic options[22]. The largest publicly accessible dataset for cancer combination drugs, such as ALMANAC, was recently published by the U.S. National Cancer Institute. This data collection contained more than 5,000 combinations of 104 investigational and licensed drugs, with synergies calculated against 60 cancer cell lines, resulting in more than 290,000 synergy scores[23]. Moreover, DrugComb (https://drugcomb.org/), a web-based portal for

storing and studying drug combination screening datasets, offers a comprehensive visualisation of drug combination susceptibility and synergy, which can significantly aid in the understanding of drug interactions at unique dosage levels. Drugcomb now has 751,498 drug combinations and 717,684 single drug screens from 37 trials, which relate to 2040 cell lines and 216 cancer forms[24].

In this work, we develop a network pharmacology approach to predict potential drug combinations for acute myeloid leukaemia (AML) based on the Beat AML dataset. We propose a drug combination strategy using bipartite network modelling of ex vivo drug screening data. By ex vivo drug response data, we directly access the individual phenotypes of the patients' cancer cells, and by network modelling, we demonstrate the similarity of drugs and AML patients. Then, we use the community structures within the drug-based bipartite networks to discover effective multi-targeted drug combination regimens (Fig. 1). Our predicted drug combinations are only suggested regarding the phenotypic interactions of the cancer cells or patient samples with the drugs without prior understanding of the genetic origin or molecular understanding of the disease.

## Results

**Defining the edge weight of bipartite networks.** In the Beat AML dataset, a set of 122 inhibitor drugs was used against 531 patient-derived AML samples. The spectra of low to high potency of drugs were observed across the patient-derived samples. However, this panel of small molecule inhibitors was selected according to their activity against the proteins involved in tyrosine and non-tyrosine kinase pathways, particularly for AML[21]. First, we determined the weight value of the drug–sample interaction to be used in the bipartite network reconstruction. This value should describe the most potent compounds for inhibiting tumour cells regarding the drug sensitivity analysis. Adding to the relative and absolute $IC_{50}$, RI value[25], and AUC, we calculated the median cell viability in the drug response experiments. The distribution of these measures was evaluated in terms of normality, skewness, and modality (Fig. 2) to choose the best measure as a weight in the bipartite network. The relationship of median to AUC was a high positive value (with the highest r Pearson correlation coefficient ~ 0.94). The distribution of medians was unimodal in contrast to $IC_{50}$ distributions, homoscedastic contrary to RI distribution, and more symmetric (non-skewed) compared to AUC distribution. In addition to investigating the linear relationship, that is, Pearson correlation analysis, we computed MIC, which measures the relationship strength, and MEV to check the closeness of the relationship to being a function. Interestingly, the relationship between median and AUC displayed higher MAS and MEV (~0.75) compared with the relationship of RI and AUC, meaning that median has a stronger association with AUC. Therefore, we have chosen the inverse of the actual median as the weight of the drug–patient interaction.

**Analysis of bipartite networks.** Furthermore, the full square submatrix (with no missing entries) of patient samples and small molecules was used as the incidence matrix of the bipartite network. Specifically, we selected the list-wise deletion strategy to remove missing values, and we used the complete cases of both variables. The downstream analysis was done on an undirected weighted bigraph comprising 176 (88 + 88) nodes and 7744 edges (Fig. 3A). The distribution of the min–max normalised edge weights indicated positive skewness, indicating that the cells were not highly sensitive to most drugs. All the performed analyses were also carried out for the GDSC dataset as proof of concept. The undirected weighted bigraph of the GDSC dataset comprised 532 (266 + 266) nodes and 70,756 edges (Fig. 3C). The distribution of the min–max normalised

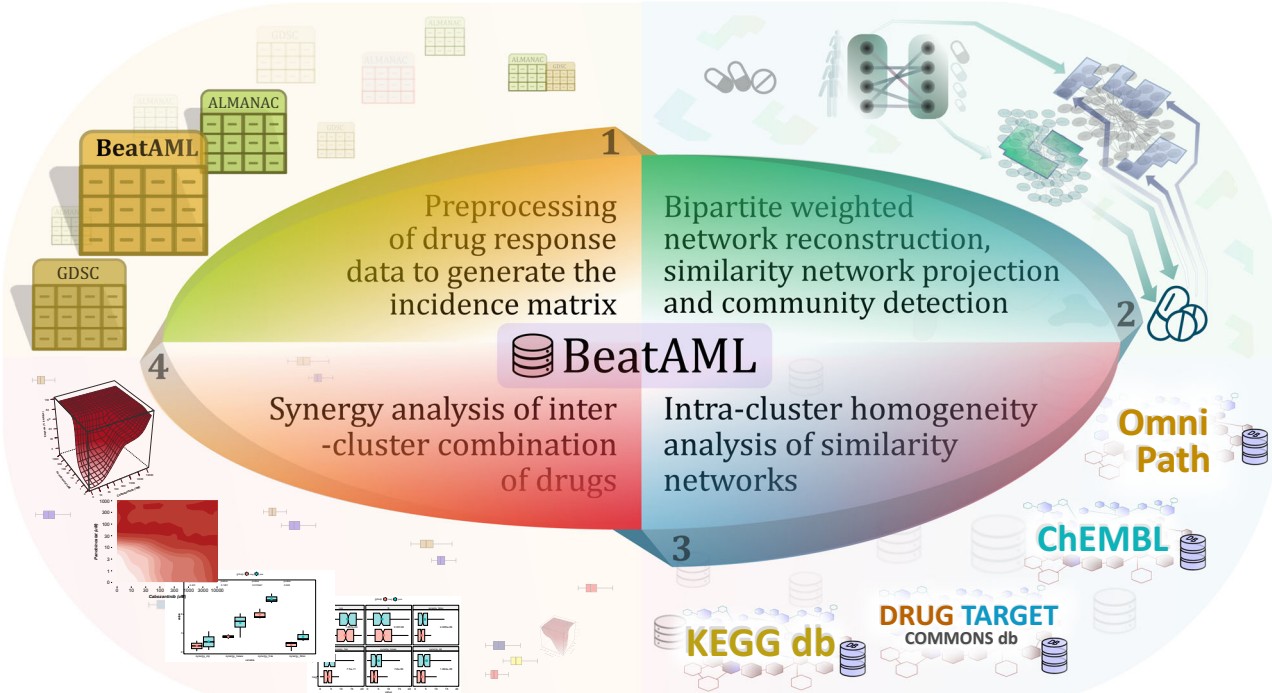

**Fig. 1 Flowchart of the study.** Data collection and pre-processing began with the Beat AML drug response dataset, which was then followed by incidence matrix extraction, weighted bipartite network reconstruction, network projection, and community detection. The intra-cluster homogeneity analysis was carried out using the similarity of drug and patient/cell members of all clusters according to available gene expression profiles, drug–target interactions, protein–protein interactions, and biological pathways using the ChEMBL, Omnipath, DrugTargetCommons, and KEGG databases. A high-throughput drug screening experiment was then used to investigate the synergistic behaviour of each drug combination that had been suggested. Furthermore, additional drug response datasets, GDSC and ALMANAC, were used for the intra-cluster homogeneity analysis and inter-cluster drug combination strategy steps, respectively, in order to examine whether the proposed workflow is dataset independent.

edge weights showed positive skewness in this dataset as well (Fig. 3D), again indicating low potency for most of the drugs. Therefore, exploring the best combination is not straightforward, and categorising drug–sample interactions seems to be required. Following the projection of these bigraphs as outlined in Fig. 3, two projected graphs, the patient similarity network (PSN) and drug similarity network (DSN), were reconstructed, such that each edge was obtained by multiplying the weighted incidence matrix. Thus, the edge weights of the projected graphs indicate the profile similarities of patient samples in PSN and small molecule inhibitors in DSN. Note that the edge weight values in DSNs and PSNs differ due to the different matrix multiplications.

The PSN and DSN of the Beat AML dataset contained 88 nodes and 3828 edges (Fig. 4), while in the GDSC projected similarity networks, there were 266 nodes and 35,378 edges. In Fig. 4, the larger the node size, the more sensitive patient-derived samples and the more potent drugs. In this subset of the Beat AML dataset, without missing data, patient 16–00627 was found to be the most sensitive and SNS-032 was the most potent inhibitor (See Supplementary Fig. 1). The community detection was subsequently done for both similarity networks via modularity score optimisation, resulting in two communities for DSN with 50 and 38 small molecules, and two communities for PSN with 39 and 49 patient samples. Alternatively, we identified two clusters of patients with distinctive drug response profiles, suggesting two subcategories of the disease. Also, we detected two clusters of small molecules, which pointed to disparate inhibiting patterns on the patient samples. In the following steps, we presented evidence of the consistency of cluster members in both networks using prior knowledge.

**Intra-cluster homogeneity analysis of similarity networks**

*Drug similarity network.* Focusing on small molecules, we presumed that inhibitory molecules with correlated effects on cell survival tended to have similar structures, purposes, and functions[26–29]. Therefore, we evaluated the similarity of SMILES structures, the analogy of protein targets, and the biological pathways of the detected clusters in the DSNs against random groupings of molecules. The distribution of the Dice similarity of the SMILES structures differed significantly between the random grouping and the clusters based on network topology (Fig. 5A). The statistical test of the median difference also resulted in the lowest $p$-values for both the pairwise two-sample Wilcoxon and Kruskal–Wallis rank sum test (adjusted $p$-value $< 2e - 16$). Evaluating their target similarities, we explored the protein targets of the small molecule inhibitors and examined the number of target intersections of small molecule pairs within the clusters. In this analysis, DTC and OmniPath were applied to explore the binding targets of small molecules and second-order node neighbours (secondary targets) in the signalling network, respectively. Assuming that proteins usually correspond to multiple signalling pathways, the KEGG database was used to check the number of pathway intersections of the protein targets for each pair of small molecules. The median similarity measures of the intersections within the network clusters significantly exceeded those for a large set of random pairs of small molecules (Fig. 5B–D) (adjusted $p$-value $< 2.2e - 16$, Kruskal–Wallis rank sum test). Comparable findings were obtained from the analysis of the GDSC dataset (Fig. 5E–H) (adjusted $p$-value $< 2.2e - 16$, Kruskal–Wallis rank sum test), suggesting that our method is also reproducible for the analysis of cell line-based datasets.

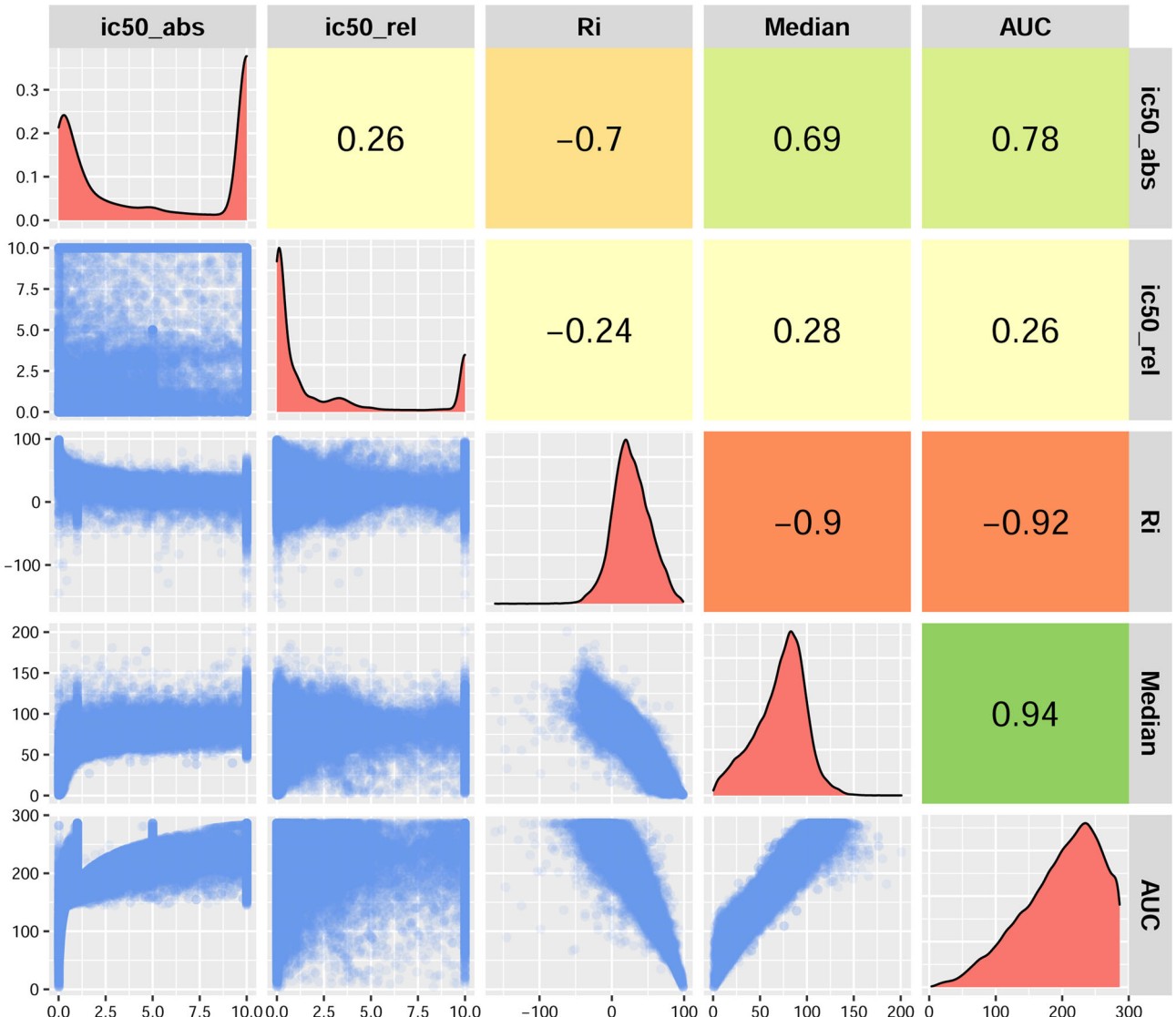

**Fig. 2 Comparison of different measures for drug response experiments in the Beat AML study.** The lower triangle of this pairwise comparison matrix shows the pairwise scatter plots for ic50_abs (Absolute IC50), ic50_rel (Relative IC50), RI (Relative Inhibition), median (the median of cell viability), and AUC (Area Under Curve of cell viability fitted line). The corresponding measures in all 10 possible pairwise associations are shown on the x- and y-axis in each scatter plot. The diagonal panel describes the histogram of each measure independently. The upper triangle represents the Pearson correlation coefficients of the corresponding pairwise comparisons. Source data are provided as a Source Data file.

*Patient and cell-line similarity network.* Next, we examined the member consistency of the patient clusters in the PSN using other available data from patient samples in the Beat AML dataset. The gene expression data, including the RPKM and CPM of the samples, were utilised to check the pairwise similarity of the cluster members. The similarity measures were also computed for a large set of random pairs of patient samples to compare with our patient stratification using network clustering. When we compared the harmonic mean similarities of the RPKM values, the pairwise similarities of patients within the clusters significantly exceeded those of the randomly selected patients (adjusted *p*-value < 2.2e − 16, Kruskal–Wallis rank sum test) (Fig. 5I). For the CPM dataset, the distributions of Jaccard distance were shown, where the distances within the clusters were statistically lower than those in the random group (adjusted *p*-value = 4.655e − 05, Kruskal–Wallis rank sum test) (Fig. 5J). For the GDSC dataset, we used the expression profiles of signature genes provided by the SPEED platform[30]. Then, differentially expressed genes were used to provide gene signatures of perturbed cancer-related pathways. In this

dataset, there were 11 activity scores to represent the activity levels of 11 well-known pathways for each cell line. Therefore, we compared the distance distributions of the cell line pairs in the clusters to a set of random pairs of cell lines. Our findings indicated that the distances within the clusters were much lower than those in the random grouping (adjusted *p*-value = 6.94e − 08, Kruskal–Wallis rank sum test) (Fig. 5K).

The Beat AML study also provided the mutational landscape in AML. Here, we used a dataset of non-benign gene mutations to characterise both clusters of patient samples. As shown in Fig. 6, both clusters of patients demonstrate a distinct profile of gene mutations regarding the involved genes and the ranks of genes based on frequency. Previously, Tyner et al. highlighted the importance of *TP53* and *ASXL* gene mutations, both responsible for the broad drug resistance patterns[21]. They further showed that mutations in certain genes may identify disease subgroups sensitive to certain inhibitors. For example, they found that patients with FLT3-ITD and *NPM1* mutations were sensitive to SYK inhibitors. Interestingly, our molecular-independent

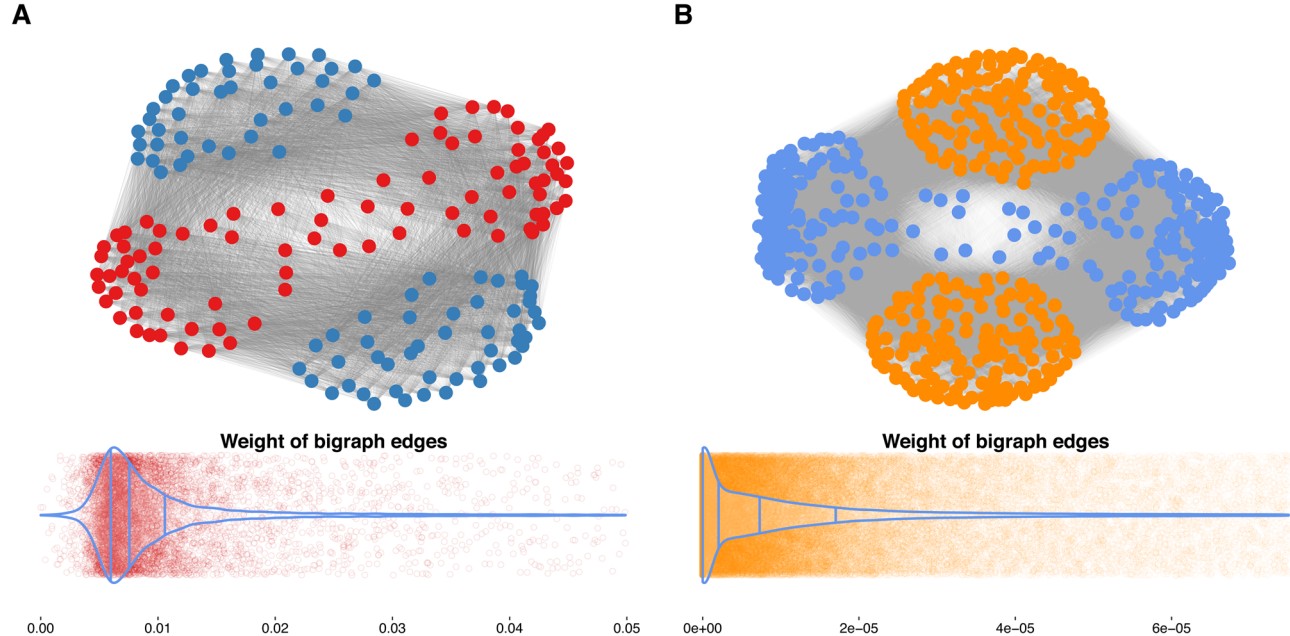

**Fig. 3 Bigraphs of cancer datasets.** The general overview of the bipartite graphs for the Beat AML (**A**) and GDSC (**B**) datasets is represented with the blue nodes as small molecule inhibitors, and red and orange nodes as patient-derived and cell line samples, respectively. The distributions of edge weight values are also depicted using violin plots together with scatter plots. From left to right, the lines in the violin plots reflect the 25th percentile, median, and 75th percentile. The colourful spots on the dim background depict how the original data was distributed. The x-axis represents edge weight, whereas the y-axis shows relative frequency. Source data are provided as a Source Data file.

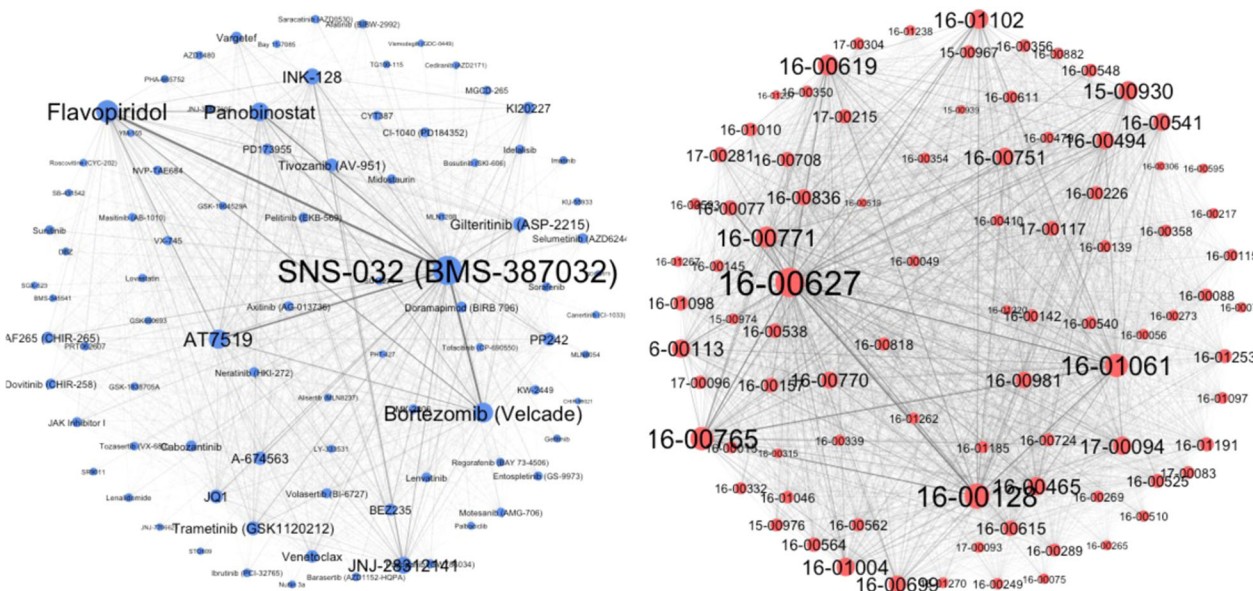

**Fig. 4 DSN and PSN of the Beat AML dataset.** The force-directed layout was selected to depict both networks. The thickness of the edges corresponds to the edge weight of the original bipartite networks after network projection, considering the weight values. The edge thickness represents the weight value of similarity between each pair of patient samples or small molecules. The node size is proportional to the strength of each node, which is the sum of the edge weights of the adjacent edges for each node. Source data are provided as a Source Data file.

network-based approach to characterise patient samples also captured the significance of the mutations above. Furthermore, our findings indicate that *TP53*, *DNMT3A*, and *NRAS* were the most frequently mutated genes in one of the patient clusters, while *TET2* and *NPM1* were the most frequently mutated genes in the other cluster, along with the FLT3-ITD mutation. These results suggest that the phenotype-level information in drug response data can corroborate the genotype-level information to stratify patients more effectively.

**Inter-cluster design strategy for drug combinations**. We assumed that the best drug combination strategy was the selection of one drug from each cluster to block potential drug resistance mechanisms and cancer recurrence. A common drug combination design could be the use of the most effective drugs of each cluster to inhibit cancer cells more effectively. However, other pharmacologic evidence can encourage the choice of the best combination of drugs more specifically. As the focus in drug combination studies also lies in finding the most synergistic drug combinations, previously

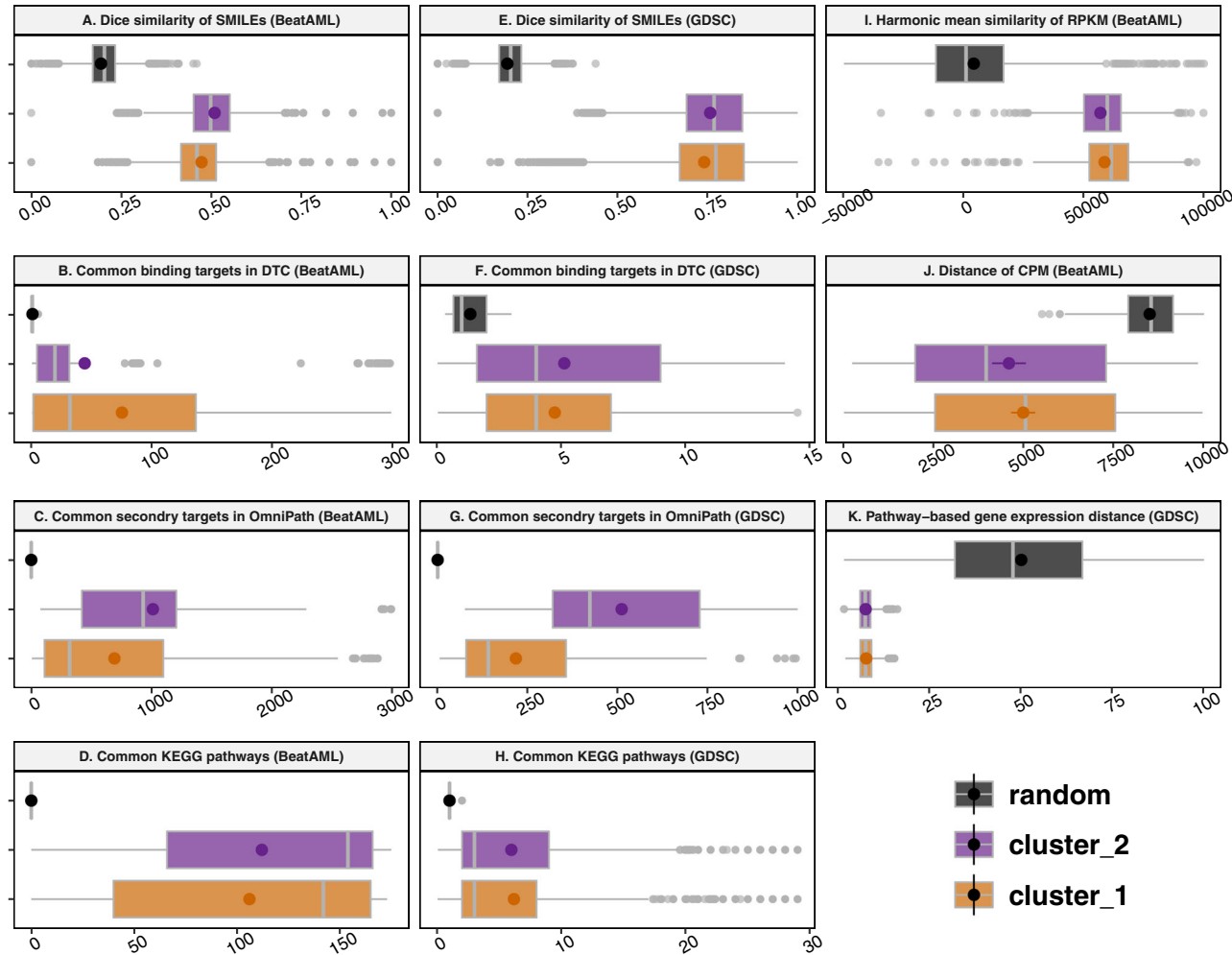

**Fig. 5 Beat AML and GDSC intra-cluster homogeneity analysis. A** The distribution of SMILE structure similarities ($n = 10,000$), **B** the distribution of pairwise intersection size of immediate binding protein targets based on the DTC database ($n = 814$), **C** secondary targets in the OmniPath database ($n = 1524$), and **D** corresponding KEGG pathways are shown for the DSN clusters compared to random grouping in Beat AML dataset ($n = 340,000$). Similarly, for the GDSC dataset, **E** the distribution of SMILE structure similarities ($n = 15,160$), **F** the distribution of pairwise intersection size of immediate protein targets based on DTC database ($n = 1257$), **G** secondary targets in the OmniPath database ($n = 4222$), and **H** associated KEGG pathways are depicted for DSN clusters compared to random grouping ($n = 17,300$). The intra-cluster homogeneity analysis was performed for PSN based on gene expression. **I** The distribution of similarity of patients' RPKM ($n = 542$), **J** the distribution of distances of patients' CPM in the Beat AML dataset ($n = 405$), and **K** the distribution of distances of pathway-based activity scores in the GDSC dataset are represented ($n = 84,485$). The coloured dots and the lines in the centre of the boxplots denote the mean and median, respectively, while the left and right hinges indicate the 25th and 75th percentiles, respectively. The left and right whiskers represent values that are no more than 1.5 times the interquartile range (IQR). The points outside the whiskers are outlier predictions. Source data are provided as a Source Data file.

reported studies were used to explore the synergy values (i.e., the degree of interactions) of drug combinations[31–34]. First, we checked if the combinations of the top five drugs (based on the median values of cell viability) of each cluster in the Beat AML and GDSC datasets (Table 1) were found in the DrugComb database. However, there were no reports regarding the 25 possible combinations of these drugs, so we aimed to compare the average synergistic values for these 10 drugs in the whole database. Figure 7 shows the distributions of synergy values in DrugComb, highlighting the mean of the synergy of the bottom and top five drugs in each network cluster. This analysis revealed the reasonably high potential of the combinations of the top five drugs according to the average median values in both Beat AML and GDSC datasets ($p$-value $= 2.96e - 02$ and $p$-value $= 3.56e - 02$, Wilcox rank sum test, respectively).

*Synergy analysis of the inter-cluster combination of drugs.* For further validation of our strategy for predicting synergistic drug

combinations using network modelling, we focused on the ALMANAC dataset[23], which has 1,892,650 combinations of 103 inhibitors tested on 60 cell lines. The same procedure as described in Fig. 1 was implemented to extract the drug modules in the DSN according to the available single drug experiments in this dataset. The median inhibition values of the single-drug responses on cell lines were used as weight values in the bipartite drug-cell line network. Using the projection of the weighted DSN, clusters of drugs with similar effect profiles on cell lines were extracted.

According to our predefined assumption, the combinations of drugs from different clusters were used as the positive group and the combinations of drugs within the clusters as the negative group. Then, we retrieved the synergy and sensitivity scores of the combinations for both groups using the DrugComb computed values, especially the highest single agent (HSA), zero-interaction potency (ZIP), Bliss, Loewe, combinational sensitivity score

(CSS), and S synergy. Figure 8A shows that the positive group of drug combinations exhibited a significantly higher value of drug synergy than the negative group. This result was evident for all types of synergy measures, indicating the superiority of the strategy of using inter-cluster drug combinations. These data also

indicate the efficiency of our proposed network-based modelling to discern drugs with similar profile effects on biological samples. Also, our proposed strategy of drug combination using the drugs of contrary clusters is more likely to acquire higher drug synergy and potency.

*High-throughput drug screening for the proposed drug combinations in AML cell lines.* To further demonstrate the ability of our model in predicting specific and robust drug combinations, experimental corroboration was conducted on a subset of 45 drug combinations for 3 AML cell lines, MOLM-16, OCI-AML3, and NOMO-1. Also, 25 out of 45 drug combinations originated from the top five drugs of the two clusters as the positive group, where higher synergy was predicted by our model, while others were the combinations of the top five drugs within each cluster, which transformed into 20 combinations as the negative group. The findings of the experimental validation of 135 drug-drug-cell line triplets are depicted in Fig. 8B using the ZIP, Bliss, HSA, and Loewe models to assess the degree of synergy. The drug combinations predicted by our model in the positive group were validated as more synergistic when considering positive scores as evidence of synergy degree (Fig. 9 and Supplementary Fig. 2). These findings were statistically more significant when using Bliss or HSA measures. These cell lines were chosen based on their genetic backgrounds and to represent a wide range of genetic variations in the Beat AML dataset's ex vivo models. We did correlation analysis of the ten (top five drugs of two clusters) selected single drug response between ex vivo model and three cell lines to illustrate the extrapolation of drug sensitivity studies in cell lines for our prediction on ex vivo models. The majority of patient-derived samples were highly correlated with these three cell lines, according to our findings (Supplementary Fig. 3). Overall, these results demonstrate the robustness of network-based predictions across various experimental setups and synergy scoring models, and the ability of our network-based model to detect new combinations of treatments.

**Fig. 6 The frequently mutated genes in the clusters of Beat AML patients.** The non-benign mutations with the probability of being damaging is greater than 0.5 were selected to find the intersection of mutated genes. The gene names are shown with the relative frequency of mutated genes in each cluster (e.g., NPM1—0.38 indicates 38% of the patients in cluster 1 have this mutation). The lines between mutated genes highlight the rank shift in the two clusters. Source data are provided as a Source Data file.

### Discussion

The availability of single-drug response datasets for cancer cell lines has prompted us to develop methods for predicting and selecting the most effective combination therapy. Several AI-based combination prediction approaches have recently been introduced that combine high-throughput molecular profiling data with drug response data to improve prediction and validation. To reflect the relationships between drug combinations, Narayan et al. used dose-response data from pharmacogenomic encyclopaedias and represented them as drug atlas[32]. Combining with the pathway/gene ontology data, their approach enables the

**Table 1 Top five small molecules in each cluster of DSNs.**

|  | Cluster 1 | Cluster 2 |
|---|---|---|
| **Beat AML dataset** | SNS-032 (BMS-387032) | Dovitinib (CHIR-258) |
|  | Flavopiridol | Nintedanib |
|  | Panobinostat | Doramapimod (BIRB 796) |
|  | AT7519 | KI20227 |
|  | Bortezomib (Velcade) | Cabozantinib |
| **GDSC dataset** | Amuvatinib | Sepantronium bromide (YM-155) |
|  | GSK690693 | Belinostat |
|  | Vinblastine | AT-7519 |
|  | AS605240 | CAY10603 |
|  | HG6-64-1 | AR-42 |

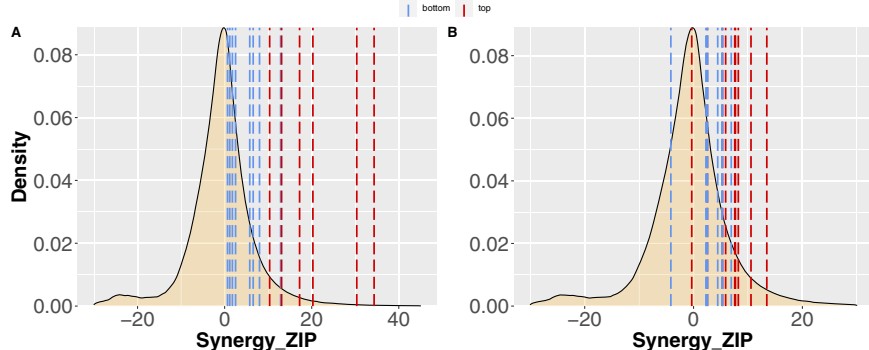

**Fig. 7 Distribution of drug combination synergy scores in the DrugComb database.** The median of synergy zip score for the top and bottom five drugs are represented by dashed lines in Beat AML dataset (**A**) and the GDSC dataset (**B**). The y-axis represents the probability density function of synergy zip in the DrugComb database, whereas the x-axis represents synergy computed using the ZIP method. Source data are provided as a Source Data file.

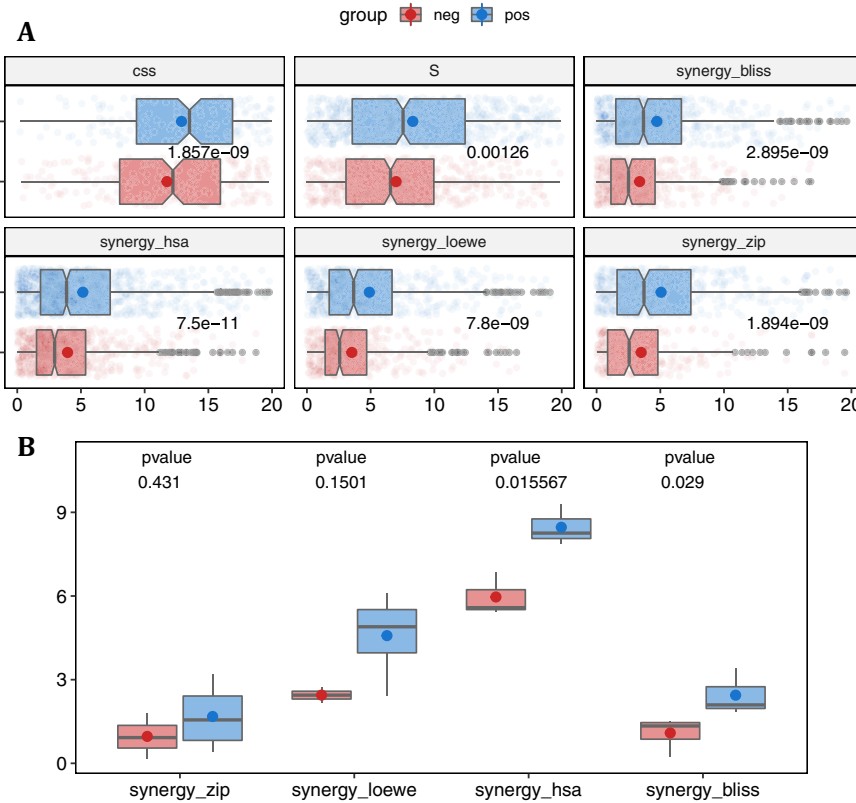

**Fig. 8 Synergy of drug combinations. A** The combinational sensitivity (CSS) and synergy scores (S, synergy_bliss, synergy_hsa, synergy_loewe, and synergy_zip) of drug combinations in the ALMANAC dataset. The top five drugs of cluster 1 (Cabazitaxel, 5-FU, Cytarabine hydrochloride, Methotrexate, Bleomycin) and cluster 2 (CHEMBL17639, Gefitinib, Ixabepilone, Dexrazoxane, Eloxatin) for inter-cluster and intra-cluster combinations are shown in blue and red as the positive and negative groups, respectively. Each plot contains a scatter plot, a notch box plot, and mean values for each group ($n = 1300$). The coloured dots and lines in the centre of the notched boxplots represent the mean and median, respectively, and the notches on the sides of the box plots can be interpreted as a comparison interval around the median values. The left and right hinges denote the 25th and 75th percentiles, respectively, while the left and right whiskers show values that are less than 1.5 times the interquartile range (IQR). Outlier predictions are shown by the points outside the whiskers, while the colourful spots in the dim background show how the original data was distributed. The p-value represents the one-sided Student's t-test significance for each score separately. **B** The measured synergy of drug combination scores in the experimental validation of selected drugs based on network modelling of the Beat AML data in three AML cell lines. Four measures of synergy, that is, ZIP, HSA, Bliss, and Loewe, are seen as box plots for the experimental confirmation of the chosen predictions ($n = 14$). Inter-cluster drug combinations are shown in blue as the positive group, and the intra-cluster combinations are shown in red as the negative group. The mean and median are shown by coloured dots and lines in the centre of the boxplots, respectively. The 25th and 75th percentiles are represented by the left and right hinges, respectively, while the left and right whiskers reflect values smaller than 1.5 times the interquartile range (IQR). The p-value represents the two-sided Student's t-test significance for each score separately. All source data are provided as a Source Data file.

prediction of combinatorial therapy, i.e., vulnerability when attacked by two drugs that can be related to tumour-driving mutations. They repeated the predicted synergies in several tumours, including glioblastoma, breast cancer, melanoma, and leukaemia mouse models, highlighting the cancer-independent prediction power of drug combination treatment. Ianevski et al. also showed that bulk viability single-agent screening assays had unexpectedly large predictability for AML cell subpopulation co-inhibition effects when combined with scRNA-seq transcriptomic data[20]. They developed a machine-learning model by combining single-cell RNA sequencing with ex vivo single-agent testing for AML with a different genetic background. They displayed an accurate prediction of synergistic patient-specific combinations while avoiding the inhibition of non-malignant cells. However, while our biomarker-independent approach relies only on the phenotypic level of information, that is, drug-response data, our predictions were compatible with the molecular profiling and biochemical annotations when it came to assessing the intra-cluster homogeneity of drugs, patients, and cell lines. Based on drug response in genetically diverse patient populations, Palmer

and Sorger, on the other hand, emphasised the independent drug action in combinatorial therapy rather than drug additivity or synergy[35]. They argued that heterogeneous responses across a population or patient-to-patient variability have a greater impact on predicting effective drug combinations. In our model, we also considered the patient's level of information when recommending drug combinations. The reconstruction of the bipartite network on a large sample of the patient population and the subsequent clustering of patients and drugs took population heterogeneity into account for drug combinations, and we also computed several synergy measures to track the synergistic behaviour of drugs rather than drug additivity.

Moreover, a training machine-learning model for predicting drug combination response, comboFM, was recently introduced using drug combination screening data as a training dataset[31]. comboFM uses a factorisation machine to model cell context-specific drug interactions through higher-order tensors. Julkunen et al. demonstrated that comboFM enables leveraging information from previous experiments performed on similar drugs and cells as training data when predicting responses of new

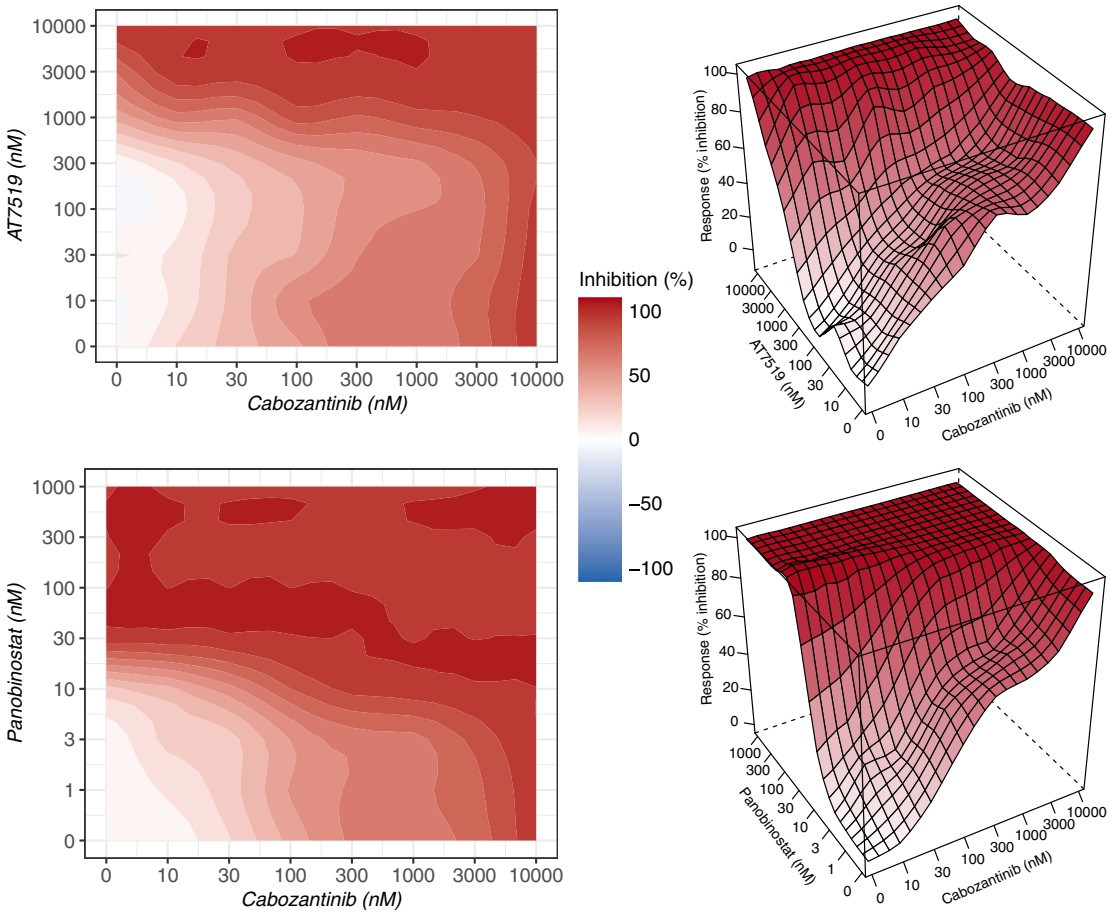

**Fig. 9 The top synergistic drug combinations identified in the positive group.** These matrices represent the highest synergistic combination based on four measures of synergy. HSA and LOEWE methods indicate that the cabozanitinb and AT7519 combination is the highest synergistic combination. The combination of cabozanitinb and panobinostat is the most synergistic based on ZIP and Bliss measures. For each combination, the sensitivity landscapes are shown in both 2D and 3D. The complete sensitivity landscapes for all 135 drug combinations can be found in Supplementary Fig. 2. Source data are provided as a Source Data file.

combinations, insofar as untested cells (testing data). They displayed high predictive performance and robust applicability of comboFM in various prediction scenarios using experimental validation of a set of previously untested drugs. However, we expounded that the prediction accuracy of the inter-cluster design strategy of drug combinations based on multipartite networks can be achieved independently from the high-quality training dataset.

Strictly speaking, in the present study, we revisited the analysis of nominal variables, namely drug name and sample identity, in drug screening results for data mining using graph theory, which we termed the nominal data mining approach[36,37]. We first considered data quality control, such as outlier detection, outlier treatment, biological and technical replicates. Because of the discrete explanatory independent variable (i.e., drug doses)[38], we assumed that regression-based measurements might even be discarded; hence, we demonstrated that median values can represent an appropriate weight score in comparing drug functionality for network reconstruction. These values were used to quantify and weight the bipartite network, which reflects the interaction strength of the drugs and biological samples. Then, two similarity networks were provided by weighted network projection to detect the topological structure of the network communities. We showed that network communities represent a rationale starting point for proposing a combinational drug regimen. Our computational and experimental validation steps amplified the logic of our proposed platform[39]. Hence, while

training datasets were not required in this method to predict drug combination, drug response data alone were adequate for the prediction, without integrating prior knowledge of biochemical profiling.

Noting that the occurrence of synergistic toxicities, which can arise from additive toxicities when targets are shared by the combined drugs, is a major barrier to applying combination therapy in the clinic[40]. If drug screening data on healthy cells are available, we suggest that a similar strategy for predicting toxicity without losing efficacy is also essential before future translational experiments. Ianevski et al. previously illustrated the importance of a desired synergy-efficacy-toxicity balance for predicting patient-customised drug combinations[20]. Hence, drug-response data on healthy cells are demanded to complement synergistic interactions of drug combinations with toxicity predictions; where drug synergy and toxicity data are optimally matched for combinatorial therapy, stronger and longer-lasting outcomes of drug combinations can be predicted. While we aimed to identify combinations with maximal synergy, we cannot discount that the effects, especially at the patient level, could be lack of toxicity rather than synergistic, and that the efficacy of the combinations may be limited to specific patient populations.

Considering these possibilities, prospective work will necessitate the provision of further patient-derived experimental validations. Despite the fact that our prediction depends solely on the drug sensitivity dataset, our suggested combinations address the

common mutational assigned aetiology of AML. This combination was proposed purely on the grounds of the phenotypic response of patient samples to the drugs, with no previous knowledge of the disease's genetic origin. In this regard, for newly diagnosed leukaemia, we recommend evaluating top combinations rather than all potential combinations (which is unfeasible) in an ex vivo drug sensitivity and resistance testing (DSRT) method reported in[41] to allow rapid translational precision medicine.

## Methods

Figure 1 presents the entire workflow of this study. The weighted bipartite network is constructed using the Beat AML dataset. This dataset is a collaborative research programme of 11 academic medical centres providing data on AML samples while offering genomics, clinical, and drug responses. It includes a cohort study of 672 tumour specimens collected from 531 patients and an analysis of 122 drug responses. To construct a weighted bipartite network, the best response read-out of drug potency was defined using information-based measures. Then, two unipartite networks were obtained using network projection on the samples and drugs. Next, communities of two projected networks were extracted, and intra-cluster homogeneity analysis was performed using the similarity of drugs and patients/cell members based on available gene expression profiles for patients, protein–protein interaction network, and biological pathways. The drug candidates for drug combination were selected from two different communities, and a high-throughput drug screening was used to assess their synergistic effects.

**Defining the response read-out for drug screening experiments**. Pharmacogenomic studies require extensive standardisation to avoid inconsistency of drug response data for further research and unbiased predictions[42,43]. Therefore, first, we controlled the quality of cell viability data to select the potent compounds. To achieve this, we examined the raw datasets regarding the availability of replicated data and outlier detection, followed by assessment of distribution, pairwise correlation, and homoscedasticity analyses to select the best response read-out or measure of drug potency. This analysis was performed using information-based nonparametric measures available in the Minerva package[44] by computing the maximal information coefficient (MIC), maximum edge value (MEV), and maximum asymmetry score (MAS). Furthermore, the relative and absolute IC50 (i.e., IC50 measures, which were computed based on the top and bottom plateaus of the curve or based on the blank and the positive control values, respectively), relative inhibition (RI) value, area under curve of drug-response fitted line (AUC), and the median of cell viability in the drug response experiments were assessed to select the best measurement. The chosen measurement was later used as a weight value for the edges of the weighted bipartite network reconstruction.

**Reconstruction and analysis of the bipartite network model**. In our bipartite network model, one group of nodes contained drugs and the other group contained cancer cell lines (in GDSC and ALMANAC) or patient samples (in Beat AML). The edges were defined by incidence matrices derived from the min–max normalised values:

$$Normalised\ value = \frac{value - minimum(values)}{maximum(values) - minimum(values)} \quad (1)$$

This normalisation transforms these values, which indicate the potency of small molecules on cancer cell lines or patient samples, into a decimal between 0 and 1. Next, we projected the bipartite network into two similarity networks: the drug similarity network and sample similarity network. In the network projection, two unipartite graphs were derived from a bipartite graph, resulting in the deduction of a similar node's relationships. In this study, we projected similarity networks that consider the edge weights in the bipartite network. Then, we studied the general properties of the networks, such as network heterogeneity, centralisation, and clustering coefficients. The critical step was community detection within the projected networks to discern functionally similar drugs and cells or patients regarding drug response. The modularity index was used to determine the best community detection algorithms, including infomap[45], fast greedy[46], and spinglass[47]. Furthermore, we explored the network modules to propose a strategy for drug combination design.

**Computational corroboration**. Multiple computational methods were applied to validate the predictions of the drug combinations and patient or cell stratification. The validation of the community structures is like the general cluster quality assessment method, and we assessed the clustering performance by matching the clustering structures to prior knowledge. This validation is foundational to possible drug combination designs. Alternatively, the combination of distinct drugs in terms of chemical structure, target profile, and implicated biological pathways is likeliest more efficient than similar drugs[7]. Therefore, we used the drug–target network, protein–protein interactions, and signalling networks to justify the similarity of cluster elements. Thus, Chembl[48], drug target commons (DTC)[49], KEGG[50], and

the OmniPath database[51] were used to extract prior annotations about the drugs and their targets. To compare the chemical structures of the drugs, a simplified molecular input line entry system (SMILES) of the drug molecules was retrieved and transformed into an extended connectivity fingerprint (ECFP) to assess the Dice similarity of the molecules. The Dice similarity is one of the standard metrics for molecular similarity calculations in which

$$S_{A,B} = 2c/(a+b) \quad (2)$$

where $a$ is the number of ON bits in molecule A, $b$ is the number of ON bits in molecule B, and $c$ is the number of ON bits in both A and B molecules[52]. Also, the corresponding gene expression profiles were used to assess similarity within a patient or cell line modules in the sample similarity networks. For reads per kilobase per million (RPKM) with negative values and counts per million (CPM), we used the Harmonic similarity and Jaccard distance, respectively, as follows:

$$S_{P,Q} = 2 \times \sum_{i=1}^{n}(P_i \times Q_i) / (P_i + Q_i) \quad (3)$$

$$D_{P,Q} = 1 - \sum_{i=1}^{n}(P_i \times Q_i) / \left(\sum_{i=1}^{n}P_i^2 + \sum_{i=1}^{n}Q_i^2 + \sum_{i=1}^{n}P_i \times Q_i\right) \quad (4)$$

where $\mathbf{P} = \{P_1, P_2, \cdots, P_n\}$ and $\mathbf{Q} = \{Q_1, Q_2, \cdots, Q_n\}$ denote the vector of gene expression values for patients or cell lines, and $n$ is the number of genes. In all cases, the similarity or distance scores were compared with the random grouping of small molecules or biological samples to perform statistical testing.

The synergy scores provided by the DrugComb database[53] were used to corroborate synergistic combinations of our network-based predictions, including HSA, Bliss, Loewe, ZIP, CSS, and S. Let us assume that drug 1 at dose $x_1$ and drug 2 at dose $x_2$ are used to produce the effects of $y_1$ and $y_2$, and $y_c$ is the effect of their combination. Drug effect is usually measured as a percentage of cell death, and a drug combination is classified as synergistic, antagonistic, or non-interactive[54]. The expected effect denoted by $y_e$ represents a non-interactive level, and it is quantified based on a reference model. Several mathematical models have been introduced to calculate the expected effect by assuming specific principles. The HSA model[55] considers the expected combination effect as the maximum of single-drug effects, that is,

$$y_e = \max(y_1, y_2) \quad (5)$$

The Loewe model[56] assumes that an individual drug produces $y_e$ at a higher dose than in the combination. In the Bliss model[57], $y_e$ is the effect of the two drugs acting independently. The ZIP model[54] considers the assumptions of the Loewe and Bliss models by assuming that, at the reference model, two drugs do not potentiate each other. CSS determines the sensitivity of a drug pair, and S synergy is based on the difference between the drug combination and the single drug dose–response curves[25].

**Cell culture and reagents**. AML cell lines MOLM-16, NOMO-1, and OCL-AML3 were purchased from DSMZ-German Collection of Microorganisms and Cell Cultures (DSMZ no. ACC 555: MOLM-16, ACC 542: NOMO-1, ACC 582: OCI-AML3). MOLM-16 and NOMO-1 were cultured in RPMI-1640 medium (Gibco/Thermo Fisher Scientific, Waltham, MA, USA) and OCI-AML3 in α-MEM (with nucleosides; Gibco/Thermo Fisher Scientific) supplemented with GlutaMAX (Gibco CTS/Thermo Fisher Scientific), foetal bovine serum (20% for MOLM-16 and OCI-AML3; 10% for NOMO-1), and antibiotics.

**Drug combination testing**. The compounds dissolved in dimethyl sulfoxide (DMSO) were plated using Beckman Coulter Echo 550 Liquid Handler (Beckman Coulter, Indianapolis, IN, USA) combined with seven concentrations for each compound in half-log dilution series with 2.5/7.5/25 nl volumes, covering a 1,000-fold concentration range on black clear-bottom TC-treated 384-well plates (Corning #3764, Corning, NY, USA). All doses were randomised across the plate to minimise any plate effects. As positive (total killing) and negative (non-effective) controls, 100 μM of benzethonium chloride and 0.2% DMSO were used, respectively.

Cells were plated on pre-administered compound plates in 25 μl (2500, 2000, or 1250 cells per well for MOLM-16, NOMO-1, and OCI-AML3 cell lines, respectively) using BioTek MultiFlo FX RAD (5 μl cassette) (Biotek, Winooski, VT, USA) and incubated for 72 h at 37 °C and 5% CO2. Cell viability was then determined by dispensing 25 μl of Cell Titre Glow 2.0 reagent (Promega, Madison, WI, USA). Plates were incubated for 5 min and centrifuged for 5 min (173 × g) before reading luminescence with a PHERAstar FS multimode plate reader (BMG Labtech, Ortenberg, Germany).

**Reporting summary**. Further information on research design is available in the Nature Research Reporting Summary linked to this article.

## Data availability

The datasets analysed during the current study are publicly available in the abovementioned repositories, i.e., Beat AML [http://vizome.org/aml/], GDSC [https://www.cancerrxgene.org/], ALMANAC [https://drugcomb.org/]. Also, the generated data in this study have been deposited in the Zenodo database under this DOI link [https://

doi.org/10.5281/zenodo.5789170]. Source data are provided with this paper. The remaining data are available in the Article and Supplementary Figures.

## Code availability

All analyses reported in this study used the statistical software R (v.4.0.0). All related R files are available in this link; https://doi.org/10.5281/zenodo.5789170.

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

## Acknowledgements

This study was financially supported by the Academy of Finland [Grant 332454 to M.J., Grant 317680 to J.T. and Grant 320131 to J.T.], and European Research Council [Grant 716063 to J.T.]. Drug screening was carried out at the FIMM High Throughput Biomedicine Unit (HTB), which is hosted by the University of Helsinki and supported by HiLIFE and Biocenter Finland. Additionally, the authors wish to acknowledge Jani Saarela and Laura Turunen of the HTB unit.

## Author contributions

M.J. and J.T. conceived of the study and supervised the project. M.J. developed the network models and led the computational analysis. M.M. and S.Z. provided computational support, while J.B. and J.E. designed and developed the experimental methods for drug sensitivity analysis. M.J., M.M., F.B., J.E. and J.T. contributed to the interpretation of the findings, and C.A.H. advised on the work. All authors contributed to the final manuscript by discussing the findings and reviewing and modifying it.

## Competing interests

C.A.H. has received research funding from BMS/Celgene, Kronos Bio, Novartis, Oncopeptides, Orion Pharma, and IMI2 projects HARMONY and HARMONYPLUS unrelated to this work. The remaining authors declare no competing interests.
