## [Peer Review File · Nature Communications]

Bipartite network models to design combination therapies in acute myeloid leukaemiaEditorial Note: Parts of this Peer Review File on page 7 have been redacted as indicated to remove third-party material where no permission to publish could be obtained.

REVIEWER COMMENTS

Reviewer #1 (Remarks to the Author): Expert in leukaemia genomics and therapy

In this study, Jafari and colleagues design a “multipartite network models” based on publicly available data sets generated with primary patient derived AML cells and cell line models to identify novel effective drug combinations.

Major comments:

- Based on the information provided in materials and methods section, readers interested in the multipartite network model will not be able to reproduce the authors' results. A more detailed description of the data that went into the analysis, as well as the algorithms and settings used is needed to make the manuscript valuable to the readers.
- As the focus of the authors is AML, it would be interesting to know whether the analysis of the GDSC and ALMANAC data sets was also only taking AML information into account? This is of importance as genes/pathways might have completely different effects in different cancers.
- Furthermore, testing so many potential combinations/potential synergistic effects will always allow the identification of “significant” results. Were the models adjusted for multiple testing?
- For many years it has been well established that distinct cytogenetic and molecular genetic aberrations are linked to drug response. In accordance, it is not surprising that clustering based on drug response does also correlate with genomic classes.
- On the other hand, it is entirely unclear how this approach could be used to predict an optimal drug combination for an individual patient. While the model can suggest novel potential synergistic combinations, how could they be assigned to a newly diagnosed leukemia. Would all combinations have to be tested in vitro?
- Finally, the examples for novel effective combinations based on studying them in three leukemia cell models are not convincing. For example as shown in Figure 9, the addition of cabozantinib does not really alter the effectiveness of Panobinostat. Panobinostat starts to effectively inhibit the respective cell line at a low nanomolar concentration of 30nM, which is not really affected by the addition of cabozantinib unless given at very high micromolar concentrations.

Minor comments:

- In the abstract the authors state the only BeatAML and GDSC were used for the bipartite models, but later in the text it is stated that also ALMANAC was used. Please clarify.
- Figure 1 is very superficial and does not really allow to grasp the multipartite network model. Furthermore, it is confusion that it mentions databased not mentioned in the text, e.g. TCGA, DEPMap.
- Many of the graphs provided are not meaningful to readers interested in AML. Axis labeling and a more detailed description might improve the value of figures.

Reviewer #3 (Remarks to the Author): Expert in drug discovery and network biology

This manuscript presented a set of network tools to identify possible drug combinations from the large-scale pharmacogenomics datasets using AML as example. Specifically, the authors integrated drug response datasets from both patient-derived ex vivo models (Beat AML) and publicly available cancer cell lines (GDSC) to build a weighted bipartite network that models the interaction of drugs and biological samples. They then prioritize possible drug combinations using a set of tools and validate several predicted cases using AML cell lines. Overall, this is an interesting and well-written, but complex methodology to identify possible drug combinations from the large-scale pharmacogenomics datasets in AML. The authors assembled various heterogeneous datasets, including chemical structures, drug-target networks, protein-protein interactions, and signaling networks, by lack of solid quality control and pre-processing. Several specific comments were listed as below for further consideration.

1. The authors integrated drug responses data from both ex vivo models and cancer cell lines. The authors are suggested to provide the correlation analyses of drug responses between ex vivo models and cancer cell lines before modeling.

2. The authors are suggested to provide more rationale how to use the median values of cell viability to compare drug potency and reconstruct a weighted bipartite network that models the interaction of drugs and biological samples. Owing to high heterogeneities of drug responses data from cancer cell lines and patient-derived ex vivo models, the median values of cell viability is highly arguable.
3. The authors assembled a set of network and bioinformatics tools in their drug combination prediction framework. Hyperparameter tuning should be provided to improve data reproducibility.
4. The authors only provided experimental validations for case studies. More comprehensive metric evaluation, such as AUC and ROC curve should be provided to illustrate the exact performance of the proposed network-based models.
5. The authors only tested community-based network models. More network topology-based measure should be tested to find the optimized network measure for drug combination prediction.

Reviewer #4 (Remarks to the Author): Expert in cancer genomics and networks, bioinformatics, and drug prediction

Authors developed a strategy for predicting drug synergy and patient stratification using multipartite networks with drug-response on patient data. They model the interaction of drugs and biological samples and leveraged the community structures to discover effective multi-targeted drug combinations and synergy levels. External validation is done on DrugComb and ALMANAC databases. They also performed wet lab validation of the potency of selective combinations of drugs against monotherapy with three AML cell lines.

1) Authors talk about "complex interactions of various biological components"

but then how do they model cell cell interactions ?

2) "For the GDSC dataset, we used the expression profiles of signature genes provided by the SPEED platform"

there is a new version of this with 14 pathways (PROGENy), you might want to take a look.

3) "previously reported studies were used to explore the synergy values (i.e. the degree of interactions) of drug combinations"

References ??

Reply to Reviewer Comments

The manuscript has been revised in response to the reviewer's comments. The following are our point-by-point replies to the reviewer's comments. In the revised manuscript, the modified text was saved as Track-changes version. We thank you and the reviewers for your thoughtful suggestions and insights, which have enhanced the quality of the manuscript and produced a more balanced and better account of the research.

Reviewer #1 (Remarks to the Author): Expert in leukaemia genomics and therapy

In this study, Jafari and colleagues design a “multipartite network models” based on publicly available data sets generated with primary patient derived AML cells and cell line models to identify novel effective drug combinations.

Major comments:

C1.1) Based on the information provided in materials and methods section, readers interested in the multipartite network model will not be able to reproduce the authors' results. A more detailed description of the data that went into the analysis, as well as the algorithms and settings used is needed to make the manuscript valuable to the readers.

R1.1) The authors would like to express their gratitude to the reviewer for drawing their attention to this issue. We provide a supplemental RMD file (Supplementary file 1) in the upgraded version that contains reconstruction and instructions for multipartite network modeling. Additionally, I'd like to inform you that we just published the NIMAA R package on the CRAN repository [1] (<https://cran.r-project.org/web/packages/NIMAA/index.html>), which includes code and programming for nominal data mining, with the BeatAML data serving as an example dataset.

C1.2) As the focus of the authors is AML, it would be interesting to know whether the analysis of the GDSC and ALMANAC data sets was also only taking AML information into account? This is of importance as genes/pathways might have completely different effects in different cancers.

R1.2) We appreciate you taking the time to comment. The main purpose of using these datasets was computational validation of our discoveries in intra-cluster homogeneity analysis and drug combination inter-cluster approach. We have not used any information of genes/pathways derived from the GDSC and ALMANAC dataset in our prediction on the BeatAML dataset. Our focus is based on patient-derived samples in the BeatAML dataset, and in order to reconstruct the bipartite network, we used the maximum number of samples with the greatest number of examined drugs, which is generally limited to a small amount of cases in cell line-based datasets. As a result, our objective was to demonstrate drug combinations based on patient-derived samples, and any prediction based on cell line data would neither validate nor reject our findings in a different set of pharmaceuticals.

C1.3) Furthermore, testing so many potential combinations/potential synergistic effects will always allow the identification of “significant” results. Were the models adjusted for multiple testing?

R1.3) We conducted statistical analysis on two conditions using a single statistical variable, e.g., "synergy_hsa" as the variable, and we compared the central measures in two conditions, positive and negative, with 25 and 20 values, respectively. To get more information, we conducted a supplementary statistical study using additional synergy metrics. We believe that multiple testing is not required in this circumstance, based on our understanding presented in [2]. However, as you can see in the supplementary file 1 (RMD file) by finding "p.adjust.method," all p-values represented in the text for Fig. 5 are adjusted p-values, and we added this word to make it more clear.

C1.4) For many years it has been well established that distinct cytogenetic and molecular genetic aberrations are linked to drug response. In accordance, it is not surprising that clustering based on drug response does also correlate with genomic classes.

R1.4) We agree with your comment. We concur that the correlation of clustering based on drug response with genomic classes is unsurprising. Indeed, if this association had not been detected, it would have been unappealing. In other words, we wanted to demonstrate that even our way of clustering, i.e., community detection within the projected similarity networks, is consistent with the genetic background, and this relationship backs up our results in terms of genetic prospective.

C1.5) On the other hand, it is entirely unclear how this approach could be used to predict an optimal drug combination for an individual patient. While the model can suggest novel potential synergistic combinations, how could they be assigned to a newly diagnosed leukemia. Would all combinations have to be tested in vitro?

R1.5) The authors would like to express their gratitude to the reviewer for bringing up this point. We also addressed this interesting point in the conclusions of the revised manuscript (Page 21). Indeed, our present work is focused on cell-based experiments, and our findings are supported by the drug combinations at this level. On the other hand, we attempted to develop a comprehensive model by including a large number of patient-derived samples in order to account for all relevant AML patient variance. As a consequence, our model provides potential synergistic combinations based on statistically significant outcomes in vitro and in silico (Fig. 8 and 9). For newly diagnosed leukemia, we advise testing top combinations rather than all possible combinations (which is impractical) in an ex vivo drug sensitivity and resistance testing (DSRT) procedure similar to that described in [3], in which 515 single drugs and multiple drug combinations were tested in four days to facilitate rapid translational medicine. Finally, a clinically effective treatment combination would be chosen for each new AML patient. However, clinical treatment plans to allocate a combination for an individual patient are beyond the scope of this manuscript, since additional clinical criteria must be addressed.

“In this regard, for newly diagnosed leukemia, we recommend evaluating top combinations rather than all potential combinations (which is unfeasible) in an ex vivo drug sensitivity and resistance testing (DSRT) method reported in (51) to allow rapid translational precision medicine.”

C1.6) Finally, the examples for novel effective combinations based on studying them in three leukemia cell models are not convincing. For example as shown in Figure 9, the addition of cabozantinib does not really alter the effectiveness of Panobinostat. Panobinostat starts to effectively inhibit the respective cell line at a low nanomolar concentration of 30nM, which is not really affected by the addition of cabozantinib unless given at very high micromolar concentrations.

R1.6) We agree with the reviewer that Panobinostat, as a non-selective histone deacetylase inhibitor, inhibits the particular cell line efficiently at a dose of 30nM. However, the plots on Fig. 9 were provided based on synergistic measures, i.e., HSA, LOEWE, ZIP and Bliss to indicate the top synergistic combinations (not manually selected). To have a better intuition, please consider the following values represented on supplementary file 3, page 39. At the preferred lower Panobinostat concentration of 10nM, adding Cabozantinib resulted in a 137 percent increase in effectiveness, while adding 30nM results in a 209 percent increase in efficacy.

Minor comments:

C1.7) In the abstract the authors state the only BeatAML and GDSC were used for the bipartite models, but later in the text it is stated that also ALMANAC was used. Please clarify.

R1.7) We appreciate your kind feedback. We updated the abstract to properly represent our process in the revised version. Take note that we utilized all three resources to create bipartite models, but with clearly defined objectives. BeatAML is the primary dataset devoted to identifying the most effective drug combinations for AML disease. GDSC dataset, which include single drug response, was used to biologically support the logic of our pipeline by intra-cluster homogeneity analysis. ALMANAC dataset was used to corroborate inter-cluster combination of drugs. ALMANAC dataset covers both single and drug combination sensitivity analysis. We began by analyzing the single drug response in this dataset and then developed bipartite models and detected communities

to predict the optimal combinations. Then, using ALMANAC combination data, we verified that our proposed drug combinations were significantly more effective. Please notice the following revised abstract:

“From the drug discovery perspective, combination therapy is recommended for cancer treatment due to its efficiency and safety compared to the common cytotoxic and single-targeted monotherapies. However, identifying effective drug combinations is time-and cost-consuming. Here, we offer a novel strategy for predicting potential drug combinations and patient subclasses by constructing multipartite networks using drug-response data on patient samples. In this study, we used Beat AML, GDSC and ALMANAC, three comprehensive datasets based on patient-derived and cell line-based samples, to show the potential of multipartite network modelling in combinatorial cancer therapy. We used the median values of cell viability to compare drug potency and reconstruct a weighted bipartite network that models the interaction of drugs and biological samples. Then, clusters of network communities were identified in two projected networks based on the topological structure of the networks. Chemical structures, drug-target networks, protein–protein interactions, and signalling networks were used to corroborate the intra-cluster homogeneity. We further leveraged the community structures within the drug-based multipartite networks to discover effective multi-targeted drug combinations and synergy levels, which were supported with more evidence using combinatorial drug responses presented in ALMANAC databases. Furthermore, we confirmed the potency of selective combinations of drugs against monotherapy in in vitro experiments using three acute myeloid leukaemia (AML) cell lines. Taken together, this study presents an innovative data-driven strategy based on multipartite networks to suggest potential drug combinations to improve AML treatment.”

C1.8) Figure 1 is very superficial and does not really allow to grasp the multipartite network model. Furthermore, it is confusion that it mentions databased not mentioned in the text, e.g. TCGA, DEPMAP.

R1.8) Thank you for sharing your viewpoint. Our graphical service team has been tasked with creating a new Figure that incorporates your suggestions. Please see the following for the new version:

C1.9) Many of the graphs provided are not meaningful to readers interested in AML. Axis labeling and a more detailed description might improve the value of figures.

R1.9) We tried our best in the current version to include your comments into all Figures and their descriptions.

Reviewer #3 (Remarks to the Author): Expert in drug discovery and network biology

This manuscript presented a set of network tools to identify possible drug combinations from the large-scale pharmacogenomics datasets using AML as example. Specifically, the authors integrated drug response datasets from both patient-derived ex vivo models (Beat AML) and publicly available cancer cell lines (GDSC) to build a weighted bipartite network that models the interaction of drugs and biological samples. They then prioritize possible drug combinations using a set of tools and validate several predicted cases using AML cell lines. Overall, this is an interesting and well-written, but complex methodology to identify possible drug combinations from the large-scale pharmacogenomics datasets in AML. The authors assembled various heterogeneous datasets, including chemical structures, drug-target networks, protein-protein interactions, and signaling networks, by lack of solid quality control and pre-processing. Several specific comments were listed as below for further consideration.

C3.1) The authors integrated drug responses data from both ex vivo models and cancer cell lines. The authors are suggested to provide the correlation analyses of drug responses between ex vivo models and cancer cell lines before modeling.

R3.1) The authors sincerely appreciate for this thoughtful suggestion. Our reasoning for corroboration analysis on cell lines is strengthened by this suggestion. The correlation analysis was carried out on both ex vivo models and cancer cell lines, and the following Figure (Supplementary file 4) was added to the manuscript with the appropriate description. As you can see, the majority of ex vivo models are highly correlated to the three cell lines, indicating that drug sensitivity analysis in these cell lines can be extrapolated to ex vivo models. In case having a limited access to patient-derived samples, we believe that this analysis could be great idea for experimental validation.

“Supplementary file 4: The correlation coefficients of ten single drug responses across ex vivo models and experimentally tested three cancer cell lines are represented in this heatmap. The color legend is displayed with the histogram of all pairwise correlation coefficients.”

“These cell lines were chosen based on their genetic backgrounds and to represent a wide range of genetic variations in the Beat AML dataset's ex vivo models. We did correlation analysis of the ten (top five drugs of two clusters) selected single drug response between ex vivo model and three cell lines to illustrate the extrapolation of drug sensitivity studies in cell lines for our prediction on ex vivo models. The majority of patient-derived samples were highly correlated with these three cell lines, according to our findings.”

C3.2) The authors are suggested to provide more rationale how to use the median values of cell viability to compare drug potency and reconstruct a weighted bipartite network that models the interaction of drugs and biological samples. Owing to high heterogeneities of drug responses data from cancer cell lines and patient-derived ex vivo models, the median values of cell viability is highly arguable.

R3.2) The heterogeneity of drug response data collected from patient-derived ex vivo models is the primary reason for avoiding statistical complex behavior such as heteroscedastic association and multimodal distribution. We aimed to choose a value that accurately described the most powerful drugs for suppressing tumor cells in terms of drug sensitivity studies. Notably, several measures make statistical assumptions about the data, such as the normality of residuals and the continuity of the explanatory variable, which are frequently violated or made more difficult in under-sampled studies examining drug responses. Leveraging commonly used drug response measurements, such as the absolute IC50, the RI value, the AUC, and the median of cell viability in drug response experiments, the distribution of these values was analyzed for normality, skewness, and modality. On the other hand, in order to select the proper measure for use as a weight in the bipartite network, we need a metric that demonstrates greater drug-drug similarity across all samples. We picked the actual median as a non-parametric central measure to describe the overall efficacy of drugs in ex vivo models. Additionally, the relationship between median and AUC was highly positive (with a Pearson correlation coefficient of 0.94 for the greatest r Pearson correlation coefficient). In comparison to IC50 distributions, the median distribution was unimodal, homoscedastic, and more symmetric (non-skewed) than the AUC distribution (Fig. 2).

C3.3) The authors assembled a set of network and bioinformatics tools in their drug combination prediction framework. Hyperparameter tuning should be provided to improve data reproducibility.

R3.3) Thank you for your comments respecting machine learning methods. We would like to highlight that we did not use hyperparameters, i.e., parameters that may be adjusted before training a model, in the bipartite network, network projection, and community detection analyses provided here. Indeed, our proposed approach is independent form training and testing approach that requires hyperparameter tuning.

C3.4) The authors only provided experimental validations for case studies. More comprehensive metric evaluation, such as AUC and ROC curve should be provided to illustrate the exact performance of the proposed network-based models.

R3.4) Note that, in this approach, we do not have hyperparameter to adjust and a binary classifier to utilize AUC (Area Under The Curve) ROC (Receiver Operating Characteristics) in order to illustrate the exact performance. Our predictions are based on community detection algorithms (clustering in network science) as an unsupervised machine learning method. Furthermore, various computational methods were also used to validate the drug, patient, and cell clustering predictions. The validation of the community structures is like the general cluster quality assessment method such as the modularity index, and we evaluated clustering performance by comparing the clustering structures to prior biological information.

C3.5) The authors only tested community-based network models. More network topology-based measure should be tested to find the optimized network measure for drug combination prediction.

R3.5) The reviewer's attention to this issue is most appreciated. As outlined in the manuscript, we assumed that the community of drugs in drug-similarity network represents functionally similar drugs and the combination of drug from distinct clusters (network communities) enhance the chance of inhibiting cell survival via distinct mode of actions. Indeed, our hypothesis for designing and conducting this study was to use community detection analysis. We would like to expand the application of other network topology-based measures for drug combination, however this will require a rationale-supported hypothesis.

Reviewer #4 (Remarks to the Author): Expert in cancer genomics and networks, bioinformatics, and drug prediction

Authors developed a strategy for predicting drug synergy and patient stratification using multipartite networks with drug-response on patient data. They model the interaction of drugs and biological samples and leveraged the community structures to discover effective multi-targeted drug combinations and synergy levels. External validation is done on DrugComb and ALMANAC databases. They also performed wet lab validation of the potency of selective combinations of drugs against monotherapy with three AML cell lines.

C4.1) Authors talk about “complex interactions of various biological components” but then how do they model cell cell interactions ?

R4.1) This phrase was used to express how the mechanism of oncopathology remained unclear for all involved gene regulatory and signaling networks. In this study, we made no attempt to model cell-cell interactions. We used the term "biochemical" instead of "biological" to avoid any misconceptions.

C4.2) “For the GDSC dataset, we used the expression profiles of signature genes provided by the SPEED platform”
there is a new version of this with 14 pathways (PROGENy), you might want to take a look.

R4.2) Thank you for your suggestion. For this analysis, we downloaded GDSC available Expression data via https://www.cancerrxgene.org/downloads/bulk_download. Based on these manually curated annotations published by GDSC database, we tried to provide computational corroborations for intra-cluster homogeneity analysis. Indeed, we did not use SPEED platform directly to provide biological pathway annotations for cell lines.

C4.3) “previously reported studies were used to explore the synergy values (i.e. the degree of interactions) of drug combinations”
References ??

R4.3) Thank you for your consideration for the dropped references. It is now resolved in the updated version.

1. Mohieddin Jafari, C.C., *NIMAA: Nominal Data Mining Analysis*. 2021.
2. Jafari, M. and N. Ansari-Pour, *Why, when and how to adjust your P values?* Cell Journal, 2019. **20**(4): p. 604-607.
3. Malani, D., et al., *Implementing a functional precision medicine tumor board for acute myeloid leukemia*. Cancer Discovery, 2021: p. candisc.0410.2021.

REVIEWERS' COMMENTS

Reviewer #1 (Remarks to the Author):

In the revised version of the manuscript the authors have addressed all my previous comments sufficiently. They have more clearly stated the scope of the manuscript, clarified many misunderstandings, improved the quality of the figures and most importantly provided additional supplementary information including a RMD file with the R codes for Beat AML bipartite network reconstruction, network projection, community detection analysis, and intra-cluster homogeneity analysis.

Reviewer #3 (Remarks to the Author):

NOTE FROM THE EDITORS: This Reviewer only provided confidential remarks to the Editor, but considered their concerns to be addressed.

Reviewer #4 (Remarks to the Author):

Nice work. all clear for me.

Reply to Reviewer Comments

The manuscript has been revised in response to the editor's comments. The following are our replies to the reviewers. In the revised manuscript, the modified text was saved as Track-changes version.

REVIEWERS' COMMENTS

Reviewer #1 (Remarks to the Author):

In the revised version of the manuscript the authors have addressed all my previous comments sufficiently. They have more clearly stated the scope of the manuscript, clarified many misunderstandings, improved the quality of the figures and most importantly provided additional supplementary information including a RMD file with the R codes for Beat AML bipartite network reconstruction, network projection, community detection analysis, and intra-cluster homogeneity analysis.

Our response: We appreciate the excellent feedback provided by the referee on our revised work, as well as all of the earlier helpful suggestions.

Reviewer #3 (Remarks to the Author):

NOTE FROM THE EDITORS: This Reviewer only provided confidential remarks to the Editor, but considered their concerns to be addressed.

Our response: We thank the referee.

Reviewer #4 (Remarks to the Author):

Nice work. all clear for me.

Our response: We thank the referee for the positive evaluation of our revised manuscript.